# Interrelating Grain Hardness Index of Wheat with Physicochemical and Structural Properties of Starch Extracted Therefrom

**DOI:** 10.3390/foods11081087

**Published:** 2022-04-09

**Authors:** Derang Ni, Fan Yang, Lin Lin, Chongde Sun, Xingqian Ye, Li Wang, Xiangli Kong

**Affiliations:** 1College of Biosystems Engineering and Food Science, Zhejiang University, Hangzhou 310058, China; derangni@163.com (D.N.); psu@zju.edu.cn (X.Y.); 2Kweichow Moutai Corporation Limited, Renhuai 564501, China; 18908520842@189.com (F.Y.); eileenjn@126.com (L.L.); 3College of Agriculture and Biotechnology, Zhejiang University, Hangzhou 310058, China; adesun2006@zju.edu.cn; 4Kweichow Moutai Group, Renhuai 564501, China

**Keywords:** grain hardness, wheat cultivars, physicochemical properties, structural properties, starch

## Abstract

To investigate the physicochemical, structural, and rheological characteristics of starch from wheat cultivars varying in grain hardness index employed in making *jiuqu* and to interrelate grain hardness index with physicochemical and structural properties of starch. Starch extracted therefrom was investigated for structural and physicochemical properties. Starch granules showed relatively wide granule size distribution; large size granules showed lenticular shapes while medium and small size granules exhibited spherical or irregular shapes. Starch from wheat with a lower grain hardness index exhibited a relatively higher degree of crystallinity. Chain-length profiles of amylopectin showed distinct differences; among the fractions of fa, fb_1_, fb_2,_ and fb_3_ representing the weight-based chain-length proportions in amylopectin, the fa fractions ranged from 19.7% to 21.6%, the fb_1_ fractions ranged from 44.4% to 45.6%, the fb_2_ fractions ranged from 16.2% to 17.0%, and the fb_3_ fractions ranged from 16.1% to 18.8%, respectively. T_o_, T_p_, T_c_, and ∆H of starch ranged from 57.8 to 59.7 °C, 61.9 to 64.2 °C, 67.4 to 69.8 °C, and 11.9 to 12.7 J/g, respectively. Peak viscosity, hot pasting viscosity, cool pasting viscosity, breakdown, and setback of starch ranged from 127 to 221 RVU, 77 to 106 RVU, 217 to 324 RVU, 44 to 116 RVU, and 137 to 218 RVU, respectively. Both G’ and G” increased in the frequency range of 0.628 to 125.6 rad/s; the wheat starch gels were more solid-like during the whole range of frequency sweep.

## 1. Introduction

Wheat is one of the oldest food crops grown in the world, and only two modern species, *Triticum aestivum* (common wheat) and *Triticum durum* (durum wheat) are grown commercially. Common wheat constitutes over 90% of the world’s wheat production. As one of the most important agricultural crops in the world, it provides approximately one third of total cereals globally [1]. Some grain characteristics, such as grain hardness, protein content, etc., determine the wheat-grain quality. Especially, grain hardness affects several processing properties and the performance of products [2]. Grain hardness is one of the primary parameters that constitute the basis of criteria of wheat quality [3]. The yield of flour, degree of starch damage, and energy inputs in the milling process are all influenced by grain hardness. The particle size, damaged starch content, and water absorption capacity of flour produced from hard wheat grains are higher than those from soft wheat grains [4]. The primary trait for identification, trade, and processing can also be distinguished by grain hardness. The hardness of grain is related to texture; soft grains are white or floury when cut and have opaque endosperm, while hard grains are horny and translucent when cut in half and have a vitreous interior. Hard wheat grains, which have a higher content of protein and increased elasticity, are necessary for bread with better quality and making pasta, as compared to soft varieties with a lower content of protein, which are desired for cakes and pastries [5,6,7]. The texture and quality of wheat grain is associated with a starch granular protein, friabilin, which constitutes the biochemical basis for assessing the texture of kernel, and two compositions, Pin a and Pin b, are present in friabilin [8]. All wheats of the soft phenotype class were observed to possess *Pin a* and *Pin b* in unaltered forms, while all hard kernel hexaploid wheats were shown to possess one (rarely two) mutation in *Pin a* or *Pin b*, most often in *Pin b* [9].

Starch composes the major proportion, approximately 80%, of wheat grains, and two major types of biomacromolecules, amylose and amylopectin, are present in starch. Amylose is an almost linear biopolymer composed of mainly α-D-(1,4)-glucosyl units with very few α-D-(1,6)-glucosyl units, while for amylopectin, the main α-D-(1,4)-glucosyl chains were attached by α-D-(1,6)-glucosyl linkages with a higher percentage of α-D-(1,6)-glucosyl units making the amylopectin much more branched. The quantity of starch in wheat grains can affect the bread staling, dough rheology, and crumb structure; furthermore, the physicochemical properties of starch granules help play a key role in the quality and yield of flour and the water retention capacity of dough; therefore, they have a great effect on the quality of the final product [10]. Coarser and damaged granules of starch were produced more easily for hard wheat cultivars than soft cultivars during the milling process, and this will affect the quality of flour practically. Granule-size distribution of starches from hard and soft wheat varieties was compared, and hard wheat was found to have a greater amount of B-type granules with a small size than soft wheat [11]. Starch isolated from hard and medium-hard wheat grains showed lower swelling power, higher amylose content, and greater crystallinity than that from extraordinarily soft wheat, and gelatinization transition temperatures of starch raised with an increase in the grain hardness index [12]. In another report, starch from the hard wheat varieties was observed to have extraordinarily higher amylose content than that from soft wheat varieties. Soft wheat cultivars showed higher starch pasting viscosity, transmittance, swelling power, and smaller crystallinity than hard wheat cultivars [5]. However, the variations in starch rheological and structural characteristics varying in wheat grain hardness were reported scarcely.

Common wheat has various applications such as bread, cakes, cookies, and noodles. In China, the main applications of wheat grains are making noodles, steamed bread, or dumpling sheets. Besides these, the wheat grains are also employed to make *jiuqu*—a fermentation starter for *baijiu* production, an ancient Chinese liquor—distinctively [13]. During *jiuqu* preparation, wheat grains are ground, mixed with water and inoculum, and then the mixture is pressed into bricks and cultured in a special room under controlled temperatures for a few weeks to form complex microbial communities with the functions of saccharification and fermentation. Grain hardness has been observed to be negatively correlated with alcohol yield previously [14]. In practice, the hardness index of wheat grains was also observed to affect the quality of *jiuqu* significantly, but the reason remained unknown. The value of wheat will increase as a specialist grain rather than a commodity, and assessment of some attributes of grain quality will favor this and guide wheat breeding and farm management. However, currently the breeding of wheat cultivars was aiming for higher yields and greater hardness of grains. To the best of our knowledge, there are few works on structural and functional properties of starch from wheat used for *jiuqu* making, and considering that the liquor production industry is an important part of the national economy in China and East Asia, the objective of the current work was to investigate the physicochemical, structural, and rheological characteristics of starch from wheat cultivars varying in grain hardness index employed in making *jiuqu*. This study would deepen our understanding of starch from various wheat grains with a different grain hardness index, and provide some theoretical information for applications of wheat grain in *jiuqu* production.

## 2. Materials and Methods

### 2.1. Materials

Eight wheat varieties (belonging to *Triticum aestivum* L.), MY2, EM596, CM37, NM2, HM8, JMX22, HG35, and JMT22, with different grain hardness indices, employed in *jiuqu* making, were collected in the current study. The wheat varieties were grown in the season of 2017–2018, and the wheat grains were harvested in 2018. MY2 and CM37 were planted in Sichuan province; EM596, HM8, and HG35 were planted in Hubei province; and NM2, JMX, and JMT22 were planted in Henan province. All the chemicals used in the current work were of reagent grade and obtained from Sigma-Aldrich (Shanghai, China).

### 2.2. Grain Hardness Index and Proximate Compositions of Wheat Grains

Wheat grain hardness index was measured according to the Chinese national standardized method of GB/T 21304-2007, which was based on the sample particle size index. Wheat grain sample with a total weight of 25 g was ground for 50 s and sieved on a wheat hardness index tester (Model JYDX 100 × 40, Hangzhou Daji Electric Instrument, China), and calculated as follows:Hardness Index (HI) = 100 − m_1_ × 100/25 − k_1_ × (12 − w) × k_2_ × (25 − t)

In the calculation, the HI of the sample was adjusted to 12% of moisture content and 25 °C of ambient temperature; m_1_, the weight of the sample that passed through the sieve after grinding; w%, the moisture content of the sample; t (°C), ambient temperature; k_1_, the correction factor of moisture; k_2_, the correction factor of temperature. The correction factors were defined according to GB/T 21304-2007.

Total starch, lipids, proteins, and fiber contents of grains from different wheat cultivars were determined using AACC standard methods [15].

### 2.3. Starch Extraction

The wheat starch was isolated according to a previous report with some modifications, which are based on the alkaline steeping method [16]. Briefly, wheat grain samples (300 g) were immersed in 900 mL 0.25% NaOH solution and blended for three minutes with an HR2168 blender (Philips, the Netherland). The mixed slurry was placed at 4 °C for 24 h, then filtered through 100 mesh (149 μm) and 400 mesh (37 μm) sieves, stepwisely. The filtrate was centrifuged at 3500× *g* for 10 min, the supernatant was discarded, and a top, yellow layer of protein was removed using a spatula. The starch sediment was resuspended with MilliQ water and neutralized by a 0.1 M HCl solution, and then centrifuged as in the above procedure. The resuspension and centrifugation were repeated for another several times until the starch sediment surface layer showed white. The purified starch was placed in an oven at 35 °C for 48 h, the dried starch samples were ground with mortar and pestle to pass through a mesh 70 (212 μm) sieve, and the starch powders were enveloped in plastic bags for further analyses.

### 2.4. Scanning Electron Microscopy (SEM)

The morphology of wheat starch granules was obtained by a scanning electron microscope (TM-1000, Hitachi, Japan). The starch samples were observed at an accelerating potential of 15 kV under low vacuum conditions, and magnified at 1000×, 2000×, and 4000×, respectively.

### 2.5. Apparent Amylose Content Determination

The iodine reagent method was employed to measure the apparent amylose content of wheat starch. Briefly, 20 mg starch sample (dry weight basis) was added by 10 mL of 0.5 M KOH to make a suspension, which was mixed by vortex. The thoroughly mixed suspension was then transferred to a 200 mL flask, and 5 mL of 1 M HCl was added, followed by iodine reagent (0.5 mL). The solution was diluted to 100 mL with MilliQ water and placed at room temperature for 20 min, and then the absorbance was measured at 620 nm on a spectrophotometer. The calculation of apparent amylose content was conducted according to a standard curve developed by using different ratios of amylose and amylopectin blends.

### 2.6. X-ray Diffraction

An X-ray diffractometer (X’Pert PRO, PANalytical, Almelo, The Netherlands) was employed to measure the X-ray diffraction pattern of wheat starches. Prior to measurements, the wheat starch powder samples were conditioned in a desiccator for 24 h. The diffractometer was operated at 40 kV and 40 mA with Cu–K radiation (1.54 Å). The diffractograms were recorded in a 2θ ranged from 5° to 45°. The step intervals were 0.02°. The peaks of crystallinity were determined on Jade software (Version 6.0, MDI, CA, USA) according to a previous publication [17], and the areas of crystalline and amorphous regions were calculated following the previous method [18]. The degree of crystallinity was calculated as follows:Degree of crystallinity (Cry%) = A_c_/(A_c_ + A_a_) × 100%
where A_c_ is the total area of ten crystalline peaks, and A_a_ is the amorphous area on the diffractograms.

### 2.7. Fourier Transform Infrared Spectroscopy (FTIR)

Nicolet 6700 FTIR spectrometer (ThermoFisher Scientific, Pittsburgh, PA, USA) with a Smart iTR diamond attenuated total reflectance (ATR) was employed to determine the short-range molecular order of wheat starch granules. The wheat starch powder samples were conditioned as described in Section 2.6 before taking measurements. The spectra were recorded at the range of 400 to 4000 cm^−1^ with a resolution of 2 cm^−1^ by 128 scans.

### 2.8. Amylopectin Chain-Length Distribution

The previous amylopectin purification method was followed to fractionate amylopectin from wheat starch. The method was based on butanol−alcohol precipitation [19]. The debranching process of purified amylopectin and determination of chain-length distributions on high performance anion-exchange chromatography (HPAEC) system (Dionex ICS-5000^+^, Sunnyvale, CA, USA) coupled with a BioLC gradient pump and a pulsed amperometric detector (PAD), were followed according to our previous description [20]. Isoamylase (from *Pseudomonas* sp., 1000 U/mL, EC 3.2.1.68, Megazyme, Bray, Co. Wicklow, Ireland) and pullulanase (from *Klebsiella planticola*, 700 U/mL, EC 3.2.1.41, Megazyme, Bray, Co. Wicklow, Ireland) were employed in the debranching reaction.

### 2.9. Thermal Properties

A differential scanning calorimeter (DSC Q100, TA Instruments, New Castle, Delaware, USA) was employed to determine the thermal properties of wheat starches. Briefly, wheat starch powder (≈2.0 mg, dry weight basis) was weighed and added into an aluminum pan, and then 6 μL of MilliQ water was added. The aluminum pan was sealed hermetically and held at room temperature for two hours. The equilibrated pan was then scanned from 35 to 110 °C at a heating rate of 10 °C/min. An empty, sealed pan was used as a reference. Thermal parameters, such as onset (T_o_), peak (T_p_), and conclusion (T_c_) temperatures and enthalpy (ΔH) of gelatinization, were calculated on the Universal Analysis 2000 (version 4.5A).

### 2.10. Pasting Properties

Rapid Visco Analyzer (RVA, model 3D, Newport Scientific, Warriewood, Australia) with Thermocline (version 2.0) was employed to determine the pasting properties. Wheat starch (3 g, 14% moisture basis) was weighed accurately in an RVA canister, and MilliQ water was added to make a total weight of 28 g and mixed thoroughly. The programmed heating and cooling cycle was used according to a previous report [21]. The starch suspension was held at 50 °C for 1 min, then heated to 95 °C at the rate of 6 °C/min, and held at 95 °C for 5 min, finally cooled to 50 °C at the same rate, and held at 50 °C for another 2 min. Peak viscosity (PV), hot paste viscosity (HPV), final (FV) and their derivative parameters breakdown (BD = PV − HPV), and setback (SB = FV − PV), were recorded and calculated, and the viscosity was expressed in Rapid Visco Units (RVU) [22].

### 2.11. Dynamic Rheological Properties

The wheat starch pastes with a concentration of 8% solids (dry weight basis, DW) were prepared by constant stirring at 95 °C in a water bath for half an hour. The prepared wheat starch pastes were transferred to the platform of a rheometer (DHR-1, TA Instruments, New Castle, DE, USA). The rheometer was equipped with a smooth 40 mm diameter parallel geometry. The gap size and strain of measurements were set at 1000 μm and 1% (within the linear viscoelastic region of these preparations, LVR), respectively.

The wheat starch pastes were cooled to 25 °C before starting the experimental temperature sweep. Wheat starch pastes were held undisturbed for 5 min at 25 °C to achieve a state of equilibrium, and then subjected to a frequency sweep at the same temperature over the range of 0.628 to 125.6 rad/s to monitor the viscoelastic functions. The sample edges were covered throughout with a thin layer of low-density silicon oil (dimethylpolysiloxane; 50 cPs viscosity) to minimize evaporation. The parameters of storage modulus (G’; solid component of the network) and loss modulus (G”; liquid component) were recorded. The tanδ is the definition of the phase angle and calculated as G”/G’.

### 2.12. Statistical Analysis

Experiments were performed using a completely randomized factorial design. All analytical determination was performed in triplicate. The data reported are an average of triplicate observations. Statistical analysis was conducted using SigmaPlot 14.0 integrated with SigmaStat 4.0 (Systat Software, San Jose, CA, USA). Results were subjected to one-way analysis of variance (ANOVA) and Tukey’s test to determine the significance of differences, and principal components analysis (PCA) was performed using XL stat to correlate and discriminate the varieties, and the result was plotted.

## 3. Results and Discussion

### 3.1. Proximate Composition of Wheat Grains

Kernel texture of wheat is an important factor for milling and flour utilization, and affects marketing and processing. The properties of kernel texture are characterized by grain hardness index, which can be measured by many methods. However, particle size index (PSI), near-infrared reflectance (NIR), and single kernel characterization system (SKCS) are the three most frequently employed. These methods rely on variation in particle-size distribution or granularity of flour or meal after milling or grinding, respectively [23,24]. The grain hardness index of collected wheat cultivar samples ranged from 37.2 (MY2) to 60.6 (JMT22), based on the PSI method (Table 1), which were lower than those of hard wheat varieties but higher than extraordinarily soft wheat varieties reported in a previous report [12]. The contents of total starch, lipid, proteins, and fiber from different wheat cultivars varying in grain hardness ranged from 78.4% for NM2 to 83.9 % for JMX22 (DW, dry weight), 1.3% for CM37 to 1.9% for MY2 and JMX22 (DW), 13.0% for JMX22 to 14.5% for NM2 (DW), and 1.9% for HM8 to 3.3% for EM596 (DW), respectively. Hardness is an extremely important genetic factor for wheat quality, and determines wheat milling behavior and hydration properties [25]. The wheat varieties exhibited lower grain weight and diameter with grain hardness index increasing. This phenomenon has been attributed to the possible compact structure of endosperm [26]. The relative hardness of wheat grains was observed to be determined by the strength of interaction between the starch granules and the protein matrix; furthermore, starch surface lipids were confirmed to be involved in the interaction of puroindolines with wheat starch, therefore affecting grain hardness [27,28]. Total starch was found to be negatively correlated with protein content; however, no significant correlations were observed between grain hardness index and parameters of proximate compositions in current work.

### 3.2. Starch Granular Morphology

Starches isolated from different wheat cultivars varying in grain hardness showed relatively wide granule size distribution. All of the large-size (>15 μm), medium-size (5–15 μm) and small-size (˂5 μm) granules were present. Large-size granules showed lenticular shapes while medium- and small-size granules exhibited spherical or irregular shapes (Figure 1). The surface of starch granules from “hard” cultivars seemed to present more grooves than that of those from soft varieties [29]. This phenomenon was also observed in our work. Figure 1 showed that starch granules from JMT22 had more grooves than those from wheat cultivars with a lower grain hardness index. A previous report found that starches from medium-hard wheat showed the largest proportion of large-size granules and the smallest proportion of medium- and small-size granules, whereas extra-soft wheat showed the greatest proportion of small-size granules [12]. Singh et al. also reported that starches from cultivars with greater hardness showed a presence of large-size granules in a larger proportion and those with a lower hardness showed a presence of medium- and small-size granules [30]. While in another report soft wheat had a higher proportion of large- and small-size granules, hard wheat had a higher proportion of medium-size granules [31].

### 3.3. Amylose Content and Structural Characteristics

Amylose content of starches from wheat cultivars varying in grain hardness showed a range from 10.4% (HG35) to 15.0% DW (JMX77) (Table 2). The variation in amylose content could be attributed to the cultivars, growth conditions, and grain hardness index [7]. The cultivars had a much greater effect than grain hardness index, even though wheat cultivars with a lower grain hardness index had lower mean amylose content [7,29,30]. Singh et al. reported amylose content between 6.2% and 20.9% DW for different wheat varieties varying in grain hardness index; however, the authors found that some hard wheat varieties had lower amylose content, and vice versa [30]. This phenomena was verified by another report [12] which was in agreement with our results, since the grain hardness index did not show any significant correlation with amylose content. Therefore, the amylose content might not play a key role in wheat grain hardness index.

The X-ray diffraction patterns of starches from wheat varying in grain hardness are illustrated in Figure 2. Relative crystallinity of starch is shown in Table 2. Wheat starch granules showed typical A-type diffraction patterns with strong diffraction peaks at 2θ of approximately 15.0°, 17.0°, 17.9°, and 23.0°, with the peaks at 17.0° and 17.9° connected together to form a dual peak. The amylose-lipid complex was characterized by a diffraction peak at ≈20.0° (Figure 2). The relative crystallinity ranged narrowly from 22.2% to 24.6%, and starch from wheat with a lower grain hardness index exhibited a slightly higher degree of crystallinity (Table 2). However, there was no significant correlation between grain hardness index and degree of crystallinity, statistically. The difference in relative crystallinity was due to amylose content and amylopectin fine structure, and higher amylose content might result in a lower degree of crystallinity [17].

ATR−FTIR is one of the most commonly used methods for measuring starch structure, and was employed as a short-range probe to determine the molecular order of starch, the absorbance at 1022 cm^−1^ seems to have a relationship with the amorphous structure, while the absorbance at 1047 cm^−1^ is sensitive to the crystalline structure [33,34]. The spectra of FTIR of starches from wheat varying in grain hardness are shown in Figure 3. Absorbance ratios of 1047 cm^−1^/1022 cm^−1^ reflecting the degree of order at the surface of starch granules were presented (Table 2), and the values of ratios ranged from 0.948 (CM37) to 0.972 (HM8). However, no significant trend was observed among wheat starches varying in wheat grain hardness.

The purified amylopectin was debranched and employed to measure the chain-length distribution parameters, which are shown in Table 2, and a graphics comparison of chain-length distribution profiles from different wheat cultivars is illustrated in Figure 4. According to a previous report, four groups of fractions, namely, fa (DP 6−12), fb_1_ (DP 13−24), fb_2_ (DP 25−36), and fb_3_ (DP > 36), were calculated to characterize the weight-based chain-length distribution of amylopectin [32]. Chain-length profiles of amylopectin from wheat cultivars varying in grain hardness index showed distinct differences: the fa fractions ranged from 19.7% to 21.6%; the fb_1_ fractions ranged from 44.4% to 45.6%; the fb2 fractions ranged from 16.2% to 17.0%; and the fb_3_ fractions ranged from 16.1% to 18.8%, respectively. However, there was no defined trend between chain-length distribution of amylopectin and grain hardness index. Starches from all wheat cultivars showed a smooth polymodal distribution with the first peak at DP 12 (DP, degree of polymerization) and the second peak at DP 45, and a shoulder at DP 18-20 was observed in all wheat starch samples (Figure 4). Similar chain-length distribution profiles were also observed in starches from normal wheat, barley, and grain amaranth [19,35,36].

### 3.4. Gelatinization Properties

T_o_, T_p_, T_c_,and ∆H of starches from different wheat cultivars varying in grain hardness index ranged from 57.8 to 59.7 °C, 61.9 to 64.2 °C, 67.4 to 69.8 °C, and 11.9 to 12.7 J/g, respectively (Table 3). Starches from cultivars with lower grain hardness were observed to show higher gelatinization transition temperatures than that from harder wheat cultivars; however, the transition temperatures decreased and enthalpies increased with decreases in grain hardness in previous reports [12,30], and ∆H of hard-wheat starch granules was found to be higher than soft-wheat granules in another report [31]. In our study, some cultivars with higher grain hardness showed higher transition temperatures and enthalpy of gelatinization (HG35); however, other cultivars with higher grain hardness showed relatively lower values (JMT22). Therefore, the relationship between grain hardness index and gelatinization properties needs further verification.

### 3.5. Pasting Properties

Pasting properties of starch from wheat cultivars varying in grain hardness are presented in Table 4. PV, HPV, FV, BD, and SB of starch from wheat cultivars varying in grain hardness ranged from 127 to 221 RVU, 77 to 106 RVU, 217 to 324 RVU, 44 to 116 RVU, and 137 to 218 RVU, respectively. Cultivar HG35, with a relatively higher grain hardness index, showed the highest values in PV, HPV, FV, BD, and SB. Cultivar EM596 and CM37, with a relatively lower grain hardness index, showed lower values in PV, HPV, FV, BD, and SB, which was in agreement with previous reports that showed starches from cultivars with a lower grain hardness usually showed lower PV and SB, and vice versa [30]. However, some soft-wheat cultivars presented higher values in pasting parameters than those of hard cultivars (Table 4). Average PV of starch from soft cultivars was higher in comparison to starch from hard varieties in a previous report [8]. Hence, the effects of grain hardness index on pasting properties were contradictory. Recently, it was reported that *Pina* expression and decreasing grain hardness, was related to increased peak and breakdown viscosity [37].

### 3.6. Dynamic Rheological Properties

Frequency sweeps over a range of 0.628 to 125.6 rad/s at 25 °C with a strain of 1% were employed to further examine the viscoelastic properties of gels formed by starch from wheat cultivars with various grain hardness indices. G’ represents the elasticity of the material and describes the energy temporarily stored in the sample that can be recovered, whereas G” is a parameter describing energy that has been used for initiation of flow and has been lost into shear heat [38]. The tanδ (G”/G’) describes the relationship between the viscous and elastic proportions of gels. The trend of values in G’ and G” of wheat starch gels dependent on frequency sweep are illustrated in Figure 5. The values in G’, G”, and tanδ at the frequency of 125.6 rad/s are presented in Table 4. Both G’ and G” increased in the frequency range of 0.628 to 125.6 rad/s. A material whose G’ and G” are frequency-dependent over a large time scale, with G’ > G”, is generally solid-like. Wheat starch gels have the values of G’ > G” without any crossover point in the range of frequency sweep, so their gels by definition have to be solid-like, and at LVR it will maintain its properties. The tanδ at the frequency of 125.6 rad/s were much lower than at 1.0, suggesting a solid-like texture, and showed a slight decrease with grain hardness index increasing with the exception of JMT22 (Table 4). G’ and G” were also observed to increase with frequency, and G’ was always larger than G”, showing a typical solid-like viscoelastic behavior for the wheat starch or flour pastes [39,40]. Starch from wheat grains with higher grain hardness index, including JMT22, showed a relatively lower storage modulus, and vice versa. JMT22 exhibited the highest proportion of longer chains (DP > 36) and intermediate amylose content. The G’ and G” of rice starch with higher amylose content was observed to have higher values than lower amylose content samples, and the values in G’ and G” were positively correlated with higher proportions of longer chains in amylopectin [41]. However, like the correlations among rheological parameters and grain hardness index, starch structural parameters were observed to be not significant (data not shown in Table 5), except that the G’ was negatively correlated with starch crystallinity (f = −0.823, *p* < 0.05).

### 3.7. Pearson and Principal Correlations among the Wheat Quality and Starch Physicochemical Properties

Pearson correlation analyses showed that TGW was negatively correlated with fa and positively correlated with fb_3_ and crystallinity, and total starch was negatively correlated with protein content and positively correlated with amylose content and fa, significantly (*p* < 0.05) (Table 5). Amylose content was observed to be negatively correlated with fb_2_ (*p* < 0.05); furthermore, fb_2_ was positively correlated with PV, FV, BD, and SB significantly (*p* < 0.01). Significant correlations among starch gelatinization temperatures and pasting parameters were also identified in the current work for wheat employed for *jiuqu* making.

Principal components analysis explains the interaction and association between the parameters by increasing the interpretability [42], and it was performed on physicochemical properties (Figure 6 and Table 6). The first four principal components could explain 87.28% of the total variance. Furthermore, the first principal component (PC 1) and the second principal component (PC 2) accounted for 32.10% and 22.59% of variations, with a cumulative 54.69% variation of the total variance (Table 6), and influence the wheat quality. PC 1 was attributed to some pasting parameters (BD, SB, PV, and FV) and fb_2_; these parameters correlated with each other significantly (Table 5). Properties such as TS and AC yielded negative contribution to the PC 1. PC 2 represented RIB, ΔH, LC, fiber, fb_3_, degree of crystallinity, etc., indicating that the inner molecular structure, including the long-rang and short-range order, was strongly associated with wheat quality, which corresponds to a previous study that crystallinity plays an important role on high amylose maize starch [43]. Hence, the results of PCA also mathematically confirms the relationship between wheat quality and total starch, amylose, crystallinity, and pasting properties with the support of Pearson’s correlation and PCA.

## 4. Conclusions

Starches isolated from different wheat cultivars varying in grain hardness showed relatively wide distribution in granule size; amylose content of starches from wheat cultivars varying in grain hardness showed a range from 10.4% (HG35) to 15.0% (JMX77). However, the amylose content might not play a key role in wheat grain hardness index. Starch from wheat with a lower grain hardness index exhibited a relatively higher degree of crystallinity. No defined trend between chain-length distribution of amylopectin and grain hardness index was observed, possibly because in the current work the grain hardness index of wheat employed in *jiuqu* making had a narrow range, or the grain hardness index of wheat has a subtle influence on amylopectin chain-length distribution. The relationship between grain hardness index and gelatinization properties needs further verification. The values of G’ wheat starch gels are always larger than G”, so the gels were more solid-like during the whole range of frequency sweep. In the current work, no significant correlations between grain hardness index and physicochemical structural properties of starch were identified. So, there is the possibility that the quality of *jiuqu* was not mainly affected by wheat starch directly, but by the interactions between starch and proteins. In the future, it will be of interest to investigate the effects of grain hardness index on the interactions between starch and proteins in wheat grains, the quality of *jiuqu*, and finally, the production of *baijiu*.

## Figures and Tables

**Figure 1 foods-11-01087-f001:**
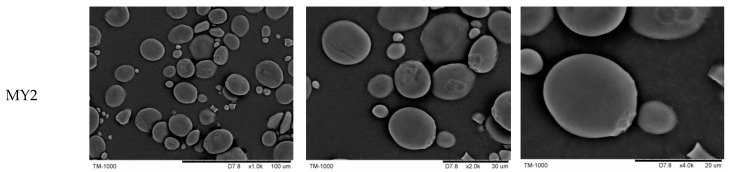
Scanning electron micrographs of starch obtained from wheat varying in grain hardness. From left to right, the starch granules were magnified at 1000×, 2000×, and 4000×, respectively.

**Figure 2 foods-11-01087-f002:**
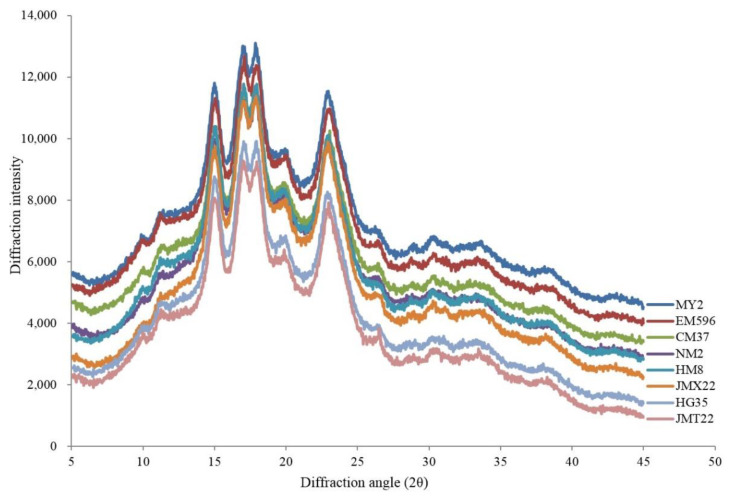
X-ray diffraction patterns of starch from wheat varying in grain hardness.

**Figure 3 foods-11-01087-f003:**
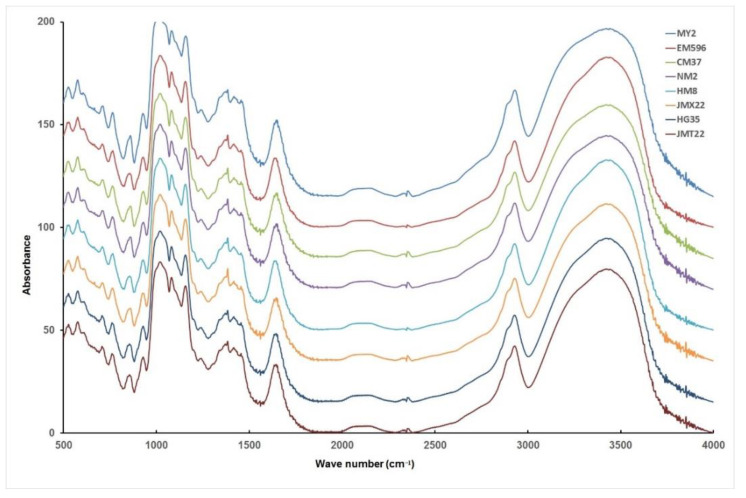
FTIR spectra of starch from wheat varying in grain hardness.

**Figure 4 foods-11-01087-f004:**
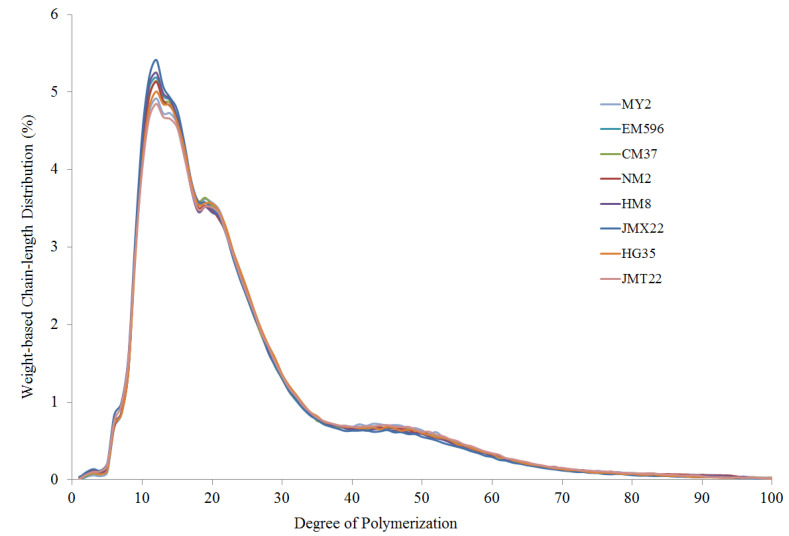
Chain-length profiles obtained by HPAEC of amylopectin from wheat varying in grain hardness.

**Figure 5 foods-11-01087-f005:**
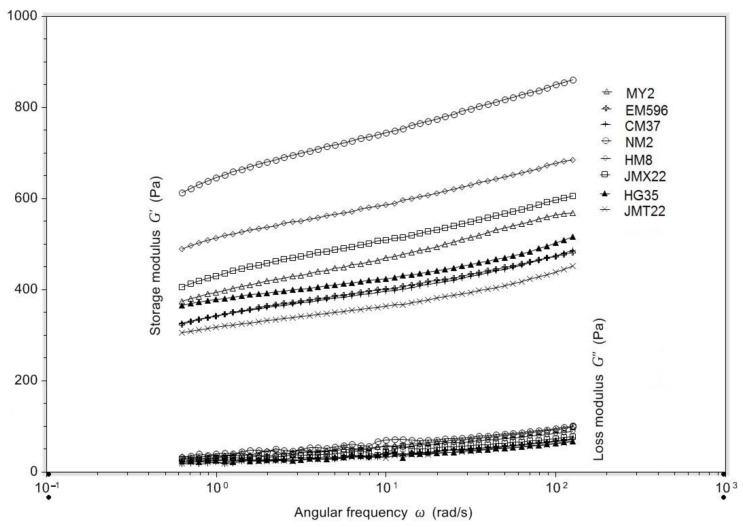
Dynamic rheological properties of starch from wheat varying in grain hardness.

**Figure 6 foods-11-01087-f006:**
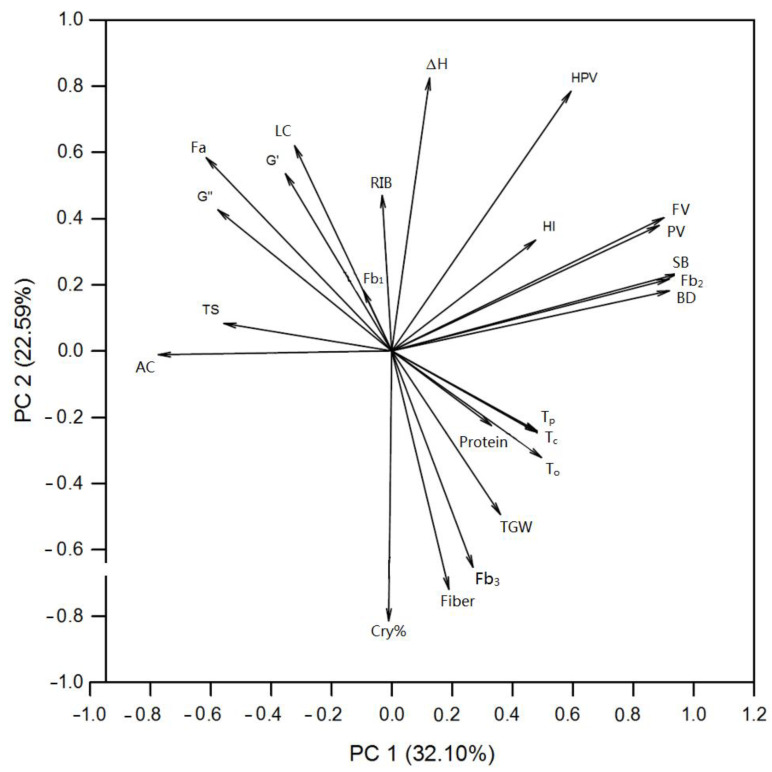
Principal components analysis of various properties of wheat varying in grain hardness index.

**Table 1 foods-11-01087-t001:** Proximate composition of wheat grains varying in grain hardness.

Cultivars	Hardness Index	Thousand Grain Weight (g)	Total Starch (%, DW)	Lipids (%, DW)	Proteins (%, DW)	Fiber (%, DW)
MY2	37.2 ± 0.3 ^e^	41.9 ± 0.6 ^a^	80.4 ± 0.9 ^a,b^	1.9 ± 0.3 ^a^	13.1 ± 0.2 ^b^	2.8 ± 0.1 ^a,b^
EM596	38.1 ± 0.2 ^e^	37.0 ± 0.3 ^b^	83.7 ± 0.5 ^a^	1.7 ± 0.0 ^a^	13.1 ± 0.1 ^b^	3.3 ± 0.0 ^a^
CM37	39.5 ± 0.5 ^e^	34.3 ± 0.3 ^c^	81.1 ± 1.1 ^a,b^	1.3 ± 0.2 ^b^	14.3 ± 0.2 ^a^	2.6 ± 0.1 ^b,c^
NM2	44.2 ± 0.1 ^d^	34.3 ± 0.2 ^c^	78.4 ± 0.3 ^b^	1.8 ± 0.1 ^a^	14.5 ± 0.1 ^a^	2.5 ± 0.2 ^b,c^
HM8	49.2 ± 0.2 ^c^	34.1 ± 0.4 ^c^	81.9 ± 1.2 ^a,b^	1.8 ± 0.1 ^a^	13.1 ± 0.5 ^b^	1.9 ± 0.0 ^c^
JMX22	53.1 ± 0.4 ^b,c^	32.6 ± 0.2 ^d^	83.9 ± 0.7 ^a^	1.9 ± 0.0 ^a^	13.0 ± 0.2 ^b^	2.2 ± 0.2 ^c^
HG35	56.0 ± 0.6 ^b^	38.0 ± 0.3 ^a,b^	79.5 ± 0.6 ^b^	1.6 ± 0.2 ^a^	13.9 ± 0.2 ^a,b^	2.8 ± 0.1 ^a,b^
JMT22	60.6 ± 0.4 ^a^	39.0 ± 0.4 ^a,b^	80.9 ± 1.0 ^a,b^	1.6 ± 0.1 ^a^	13.4 ± 0.1 ^b^	2.6 ± 0.3 ^b,c^

Values are shown as mean ± standard deviation of three replicates and means in each column followed by different letters are significantly different (*p* < 0.05).

**Table 2 foods-11-01087-t002:** Amylose content, crystalline property, and amylopectin weight-based chain-length distribution of starch from wheat varying in grain hardness index.

Cultivars	Amylose (g/100 g DW)	fa DP6–12 (%)	fb_1_ DP13–24 (%)	fb_2_ DP25–36 (%)	fb_3_ DP > 36 (%)	Crystallinity (%)	Ratios of Infrared Band (1047 cm^−1^/1022 cm^−1^)
MY2	12.7 ± 0.1 ^b^	19.7 ± 0.2 ^b^	45.2 ± 0.3 ^a,b^	16.3 ± 0.2 ^b^	18.6 ± 0.1 ^a,b^	24.6 ± 0.2 ^a^	0.967 ± 0.005 ^a,b^
EM596	14.3 ± 0.3 ^a,b^	20.5 ± 0.1 ^a^	45.5 ± 0.2 ^a^	16.2 ± 0.2 ^b^	17.4 ± 0.1 ^b^	24.0 ± 0.1 ^a,b^	0.954 ± 0.002 ^b,c^
CM37	12.0 ± 0.0 ^b,c^	20.2 ± 0.4 ^a,b^	45.4 ± 0.5 ^a^	16.2 ± 0.4 ^b^	17.8 ± 0.3 ^b^	24.1 ± 0.3 ^a,b^	0.948 ± 0.003 ^c^
NM2	12.1 ± 0.2 ^b,c^	20.2 ± 0.2 ^a,b^	44.9 ± 0.1 ^a,b^	16.5 ± 0.2 ^a,b^	17.9 ± 0.3 ^b^	22.2 ± 0.2 ^c^	0.954 ± 0.006 ^b,c^
HM8	11.7 ± 0.1 ^c^	20.7 ± 0.0 ^a^	45.0 ± 0.1 ^a,b^	16.5 ± 0.2 ^a,b^	17.3 ± 0.2 ^b^	22.8 ± 0.2 ^b^	0.972 ± 0.002 ^a^
JMX22	15.0 ± 0.4 ^a^	21.6 ± 0.3 ^a^	45.6 ± 0.4 ^a^	16.2 ± 0.3 ^b^	16.1 ± 0.2 ^c^	22.6 ± 0.1 ^b,c^	0.953 ± 0.010 ^b,c^
HG35	10.4 ± 0.1 ^d^	19.7 ± 0.1 ^b^	45.4 ± 0.2 ^a^	17.0 ± 0.1 ^a^	17.6 ± 0.0 ^b^	23.1 ± 0.3 ^b^	0.956 ± 0.004 ^b^
JMT22	12.6 ± 0.2 ^b^	19.7 ± 0.0 ^b^	44.4 ± 0.1 ^b^	16.6 ± 0.2 ^a,b^	18.8 ± 0.1 ^a^	24.4 ± 0.2 ^a^	0.953 ± 0.008 ^b,c^

Values are shown as mean ± standard deviation of three replicates and means in each column followed by different letters are significantly different (*p* < 0.05). DP, degree of polymerization; fa (DP 6–12), fb_1_ (DP 13–24), fb_2_ (DP 25–36), and fb_3_ (DP > 36) are defined according to the division suggested by a previous report [32].

**Table 3 foods-11-01087-t003:** Thermal properties of starch from wheat varying in grain hardness index.

Cultivars	T_o_ (°C)	T_p_ (°C)	T_c_ (°C)	∆H (J/g)
MY2	57.8 ± 0.2 ^b^	61.9 ± 0.1 ^c^	67.4 ± 0.3 ^b^	12.6 ± 0.1 ^a^
EM596	59.3 ± 0.1 ^a^	63.6 ± 0.2 ^a,b^	69.5 ± 0.2 ^a^	12.3 ± 0.2 ^a,b^
CM37	59.2 ± 0.0 ^a^	63.8 ± 0.1 ^a,b^	69.0 ± 0.4 ^a,b^	12.1 ± 0.0 ^b^
NM2	59.3 ± 0.3 ^a^	63.3 ± 0.3 ^b^	68.0 ± 0.1 ^b^	12.4 ± 0.3 ^a,b^
HM8	57.9 ± 0.1 ^b^	62.2 ± 0.2 ^c^	67.8 ± 0.2 ^b^	12.7 ± 0.1 ^a^
JMX22	58.1 ± 0.2 ^b^	62.6 ± 0.1 ^b,c^	68.4 ± 0.1 ^b^	12.7 ± 0.4 ^a^
HG35	59.7 ± 0.4 ^a^	64.2 ± 0.1 ^a^	69.8 ± 0.4 ^a^	12.7 ± 0.2 ^a^
JMT22	58.4 ± 0.1 ^b^	62.4 ± 0.2 ^c^	68.2 ± 0.5 ^b^	11.9 ± 0.1 ^b^

Values are shown as mean ± standard deviation of three replicates and means in each column followed by different letters are significantly different (*p* < 0.05). T_o_, T_p_, T_c_ represent peak, onset, and conclusion temperatures, respectively, and ΔH represents enthalpy of gelatinization.

**Table 4 foods-11-01087-t004:** Pasting and rheological properties of starch from wheat varying in grain hardness index.

Cultivars	PV (RVU)	HPV (RVU)	FV (RVU)	BD (RVU)	SB (RVU)	G’ (Pa)	G” (Pa)	tanδ
MY2	156 ± 2 ^b^	88 ± 1 ^b^	229 ± 4 ^c^	68 ± 1 ^b^	141 ± 6 ^b,c^	432 ± 6 ^c,d^	70 ± 3 ^c,d^	0.16 ± 0.01 ^a^
EM596	129 ± 3 ^d^	77 ± 3 ^c^	217 ± 6 ^d^	52 ± 0 ^d^	140 ± 1 ^b,c^	488 ± 8 ^c,d^	74 ± 1 ^c^	0.15 ± 0.01 ^a,b^
CM37	127 ± 1 ^d^	83 ± 1 ^c^	219 ± 2 ^d^	44 ± 3 ^e^	137 ± 3 ^c^	489 ± 15 ^c^	69 ± 1 ^c,d^	0.14 ± 0.00 ^b,c^
NM2	138 ± 2 ^c^	91 ± 2 ^b^	234 ± 2 ^b,c^	47 ± 1 ^e^	142 ± 4 ^b,c^	891 ± 32 ^a^	106 ± 7 ^a^	0.12 ± 0.02 ^c^
HM8	160 ± 6 ^b^	99 ± 4 ^a,b^	249 ± 4 ^b^	61 ± 2 ^c^	150 ± 2 ^b^	649 ± 31 ^b^	87 ± 4 ^b^	0.13 ± 0.01 ^c^
JMX22	148 ± 4 ^b,c^	95 ± 3 ^a,b^	241 ± 3 ^b^	53 ± 2 ^d^	146 ± 2 ^b^	612 ± 9 ^b^	79 ± 2 ^b,c^	0.13 ± 0.01 ^c^
HG35	221 ± 5 ^a^	106 ± 4 ^a^	324 ± 5 ^a^	116 ± 3 ^a^	218 ± 1 ^a^	430 ± 25 ^d^	54 ± 3 ^d^	0.13 ± 0.01 ^c^
JMT22	151 ± 2 ^b,c^	85 ± 2 ^b,c^	240 ± 2 ^b^	65 ± 2 ^b,c^	155 ± 3 ^b^	382 ± 19 ^d^	62 ± 6 ^d^	0.16 ± 0.02 ^a,b^

Values are shown as mean ± standard deviation of three replicates and means in each column followed by different letters are significantly different (*p* < 0.05). PV, peak viscosity; HPV, hot paste viscosity; FV, final viscosity; BD, break down; SB, setback. G’ and G” are values at the frequency of 125.6 rad/s, and tanδ is calculated as G”/G’ at the frequency of 125.6 rad/s.

**Table 5 foods-11-01087-t005:** Pearson correlation coefficients among physicochemical and structural properties of wheat starch and grain hardness index.

	HI	TGW	TS	LC	PC	FC	AC	fa	fb_1_	fb_2_	fb_3_	Cry%	RIB	T_o_	T_p_	T_c_	∆H	PV	HPV	FV	BD	SB	G’
TGW	−0.08																						
TS	−0.06	−0.29																					
LC	−0.01	0.07	0.15																				
PC	−0.08	−0.23	−0.73	−0.56																			
FC	−0.39	0.57	0.01	−0.24	0.06																		
AC	−0.19	−0.21	0.78	0.40	−0.56	0.11																	
fa	−0.01	−0.78	0.72	0.36	−0.37	−0.47	0.67																
fb_1_	−0.42	−0.28	0.46	0.02	−0.13	0.22	0.31	0.48															
fb_2_	0.64	0.26	−0.60	−0.10	0.28	−0.05	−0.77	−0.53	−0.34														
fb_3_	−0.08	0.76	−0.61	−0.26	0.24	0.35	−0.45	−0.91	−0.69	0.26													
Cry%	−0.08	0.73	0.12	−0.35	−0.25	0.55	0.07	−0.54	−0.15	−0.23	0.64												
RIB	−0.10	0.27	−0.03	0.58	−0.50	−0.38	−0.22	−0.04	−0.15	0.13	0.10	−0.04											
T_o_	−0.05	−0.12	−0.33	−0.57	0.69	0.54	−0.34	−0.29	0.23	0.35	0.01	−0.16	−0.63										
T_p_	−0.06	−0.23	−0.18	−0.62	0.63	0.46	−0.30	−0.14	0.42	0.28	−0.17	−0.17	−0.64	0.96									
T_c_	0.10	−0.12	0.16	−0.56	0.23	0.53	−0.11	−0.04	0.49	0.26	−0.28	0.01	−0.57	0.81	0.89								
∆H	−0.03	−0.15	0.07	0.63	−0.30	−0.33	−0.06	0.37	0.51	0.17	−0.56	−0.54	0.58	−0.23	−0.12	−0.10							
PV	0.54	0.30	−0.36	0.06	−0.01	−0.05	−0.61	−0.34	0.05	0.87	0.00	−0.18	0.27	0.17	0.21	0.30	0.51						
HPV	0.54	−0.16	−0.31	0.25	0.05	−0.54	−0.52	0.07	0.07	0.70	−0.31	−0.60	0.38	−0.03	0.04	0.01	0.72	0.83					
FV	0.60	0.12	−0.36	−0.04	0.13	−0.07	−0.62	−0.26	0.07	0.89	−0.10	−0.32	0.11	0.33	0.37	0.42	0.46	0.97	0.84				
BD	0.48	0.45	−0.34	−0.02	−0.02	0.16	−0.59	−0.46	0.05	0.85	0.12	0.00	0.19	0.25	0.27	0.39	0.39	0.97	0.68	0.93			
SB	0.57	0.22	−0.35	−0.15	0.15	0.10	−0.61	−0.35	0.08	0.89	−0.02	−0.18	0.00	0.43	0.46	0.54	0.33	0.95	0.72	0.98	0.96		
G’	−0.15	−0.65	−0.23	0.37	0.38	−0.45	0.04	0.43	−0.04	−0.11	−0.35	−0.82	0.06	0.07	−0.00	−0.32	0.28	−0.28	0.18	−0.18	−0.44	−0.30	
G”	−0.30	−0.55	−0.10	0.46	0.20	−0.41	0.20	0.44	−0.11	−0.32	−0.25	−0.65	0.18	−0.13	−0.21	−0.49	0.19	−0.49	−0.04	−0.43	−0.62	−0.54	0.95

HI, Hardness Index; TGW, Thousand Grain Weight; TS, Total Starch; LC, Lipid Content; PC, Protein Content; FC, Fiber Content; AC, Amylose Content; Cry%, Crystallinity (%); RIB, Ratios of infrared band (1047 cm^−1^/1022 cm^−1^); ■■■ means positive correlations at *p* < 0.05, *p* < 0.01 and *p* < 0.001 levels, respectively; ■■ means negative correlations at *p* < 0.05, *p* < 0.01 levels, respectively.

**Table 6 foods-11-01087-t006:** Eigenvector values for the first five principal components.

	PC 1	PC 2	PC 3	PC 4	PC 5
HI	0.172	0.144	−0.057	0.021	−0.576
TGW	0.130	−0.212	−0.322	0.105	0.226
TS	−0.200	0.036	0.073	0.404	−0.176
LC	−0.116	0.266	−0.203	0.062	0.231
Protein	0.119	−0.097	0.237	−0.393	0.054
Fiber	0.068	−0.309	0.079	0.166	0.320
AC	−0.279	−0.005	0.055	0.252	−0.077
Fa	−0.221	0.250	0.189	0.161	−0.145
Fb_1_	−0.034	0.079	0.268	0.320	0.341
Fb_2_	0.331	0.093	−0.059	−0.103	−0.088
Fb_3_	0.097	−0.281	−0.274	−0.218	0.050
Cry%	−0.004	−0.350	−0.208	0.182	0.003
RIB	−0.012	0.201	−0.335	0.013	0.258
T_o_	0.179	−0.138	0.353	−0.090	0.127
T_p_	0.174	−0.104	0.395	−0.004	0.102
T_c_	0.174	−0.106	0.339	0.217	−0.011
ΔH	0.045	0.354	−0.023	0.144	0.355
PV	0.319	0.163	−0.075	0.109	0.041
HPV	0.214	0.336	−0.028	−0.018	−0.017
FV	0.325	0.173	0.020	0.072	−0.023
BD	0.332	0.078	−0.081	0.152	0.070
SB	0.341	0.100	0.036	0.103	−0.027
G’	−0.127	0.230	0.153	−0.350	0.137
G”	−0.208	0.183	0.073	−0.326	0.168
Proportion of Variance Interpretation (%)	32.100	22.590	18.310	14.280	7.310

## Data Availability

Data is contained within the article.

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
