# Peer review of "Interrelating Grain Hardness Index of Wheat with Physicochemical and Structural Properties of Starch Extracted Therefrom"

_foods, 2022, doi:10.3390/foods11081087_

Round 1

Reviewer 1 Report

Compared to the previous version, the present form of the manuscript appears to be improved considerably in terms of title, technical content, and overall presentation. Accordingly, I recommend acceptance of the manuscript in its present form, and can be processed further for publication. 

Author Response

We thank you so much again for your nice suggestions and comments on our manuscript to improve the quality.

Reviewer 2 Report

The undertaken topic by the authors is quite interesting. The justification for selected topic is quite clear, however more sources could be included. Especially considering that wheat is quite commonly investigated product. The methods need to be described with more detail. Moreover, I don’t see the correlation between conducted experiment and usability in terms jiuqu production.

Line 18-19 the fb fractions are not clear for the reader at this moment

Line 22 just breakdown and setback (as it is not directly viscosity value, its difference)

Line 43-36 style

Line 47 is associated (style)

Line 54 “having few branches connected by α-D-(1,6)-glucosyl linkage” should be deleted it adds unnesesarly too much complexity to the sentence and to unfamiliar reader this sentence will be hard to understand/misleading.

Line 56 density is not a proper term (in this case).

Line 88-89 repetition of information.

Line 98-100 more information on the material should be provided, how they were obtain, year of cultivation, region (if possible)

Line 105 how the grinding was sieving preformed?

Line 134-142 why the determination was performed in suspension?

Line 144-153 were the samples conditioned prior the measurement?

Line 155-158 samples were in powdered from or in KBr pellet? Were the samples conditioned prior the measurement?

Line 160 what previous methods?

Line 184 provide some basic information, heating cooling rate, temperature etc.

Line 185 please use term final viscosity (it is more common).

Line 203 I don’t quite understand how the experiment was factorial.

Line 246 the trimodal statement has to be backed by the analysis of granule size distribution.

Line 271-272 the cause should be presented in opposite manner.

Line 284 this statement is debatable, IR is starch has some niche applications.

Line 334-348 I think it would be beneficial to include pasting curves.

Line 360 I don’t quite undesrand why G’ and G’’ values were presented in table as they are also visible in the graph. Morover the value of phase angle tangent is not clear (to what angular velocity it is related, moreover it is not discussed).

Line 367 – quality is not a proper term in this case.

Line 368 – viscosity is represented by complex viscosity value in this case, whereas G’’ is parameter describing energy that has been used for initiation of flow and has been lost into shear heat.  This is well presented for example in https://www.mdpi.com/2304-8158/8/7/240/htm

Line 374 gel by definition has to be solid like and at LVR it will maintain its properties.

Line 392 drawing conclusions prom PCA plot explaining only 55% of variance is “risky”.

Line 386-401 it would be better if this paragraph would focus more on person correlation coefficients.

Reviewer 3 Report

This paper presented for the review is dedicated to the investigation of the physicochemical, structural, and rheologic properties of starch from wheat cultivars, employed in production of juiqui, fermentation starter for traditional liquour baijiu in China, and correlations ot these properties with grain hardness index. The paper contains comprehensive analyzes and represents a great contribution in the field of research of wheat grain in juigu production. But some issues need to be clarified.

The abstract lacks the part that is described in the title. Consider the title and main objectives of the paper in the abstract section.

In the introduction section, the authors gave enough information on the subject area of article but need to cover with more references.

Line 214-218, Why was the PSI method chosen to measure the grain hardness index?

Line 220-222, Write for which wheat cultivars  are the largest and for which the smallest values.

What conclusion obtained of lines 273-278?

Line 346, „PINA“? Did you mean on protein puroindoline, which mentioned in the introduction and marked as Pina , write to be unique

Table 5. Why parameters G' and G'' were not included in correlation analysis?

Line 390-392,  The first two component, PC1 and PC2 accounted for 54.69% of the total variance. Is that sufficient for data representation? Consider projection of variables PC1 and PC3, because the complete contribution of PC1 and PC2 of the total variance is not higher than 70%.

What is wrong with the page numbering?

Round 2

Reviewer 2 Report

All comments have been addressed and suggestions addressed correctly, the submitted manuscript can be recommended for publication.

This manuscript is a resubmission of an earlier submission. The following is a list of the peer review reports and author responses from that submission.

Round 1

Reviewer 1 Report

The present manuscript envisages detailed inter-relations among various parameters including grain hardness index, relative proportion of protein/ starch/ lipid/ fiber, amylose/ amylopectin content, gelatinization temperature, and dynamic rheological properties. Authors have drawn some interesting conclusions, e.g. the minimum contribution of amylose content in determining the wheat grain hardness index. Moreover, starch extracted from wheat with the lower grain hardness index exhibited the higher degree of crystallinity. At the same time, there is no clear outcome regarding relationship between grain hardness index and gelatinization properties. In addition, the rheological properties alongside relationships among storage and loss moduli appear to be less detailed. In my opinion, to improve further the overall presentation, authors should modify their manuscript based on the following comments:

  1. Title: The title needs modification to make it grammatically correct as well as to make it more acceptable to the readers. The modified title can be ‘Interrelating grain hardness index of wheat with physicochemical and structural properties of starch extracted therefrom’.

  1. In my opinion, to generate more interests in readers, the as-prepared products should be employed in making jiuqu followed by comparing the ease or difficulty in manufacturing the jiuqu.

  1. Authors should mention the scientific names of the species for all the eight varieties of wheats.

  1. In page 4 of 17, authors have stated the following: ‘The relative hardness of wheat grains was observed to be determined by the strength of interaction between the starch granules and the protein matrix’. This means that the starch granules function as reinforcing filler in the surrounding protein matrix. In fact, such reinforcing effect is closely associated with the conventional and non-conventional hydrogen bonding among starch and protein matrix. Effects of such hydrogen bonding can be visualized via changing the nature of FTIR peaks. In this regard, authors may go through the following papers: ACS Omega 2019, 4, 1, 1763–1780; Macromolecular Materials and Engineering, 295, 1025–1030; DOI: 10.1016/B978-0-12-823791-5.00006-5.

  1. The FTIR plots should be depicted in the revised manuscript. Comparing these plots with a starch specific plot can be an effective strategy to investigate the nature of hydrogen bonds in between starch and protein.

  1. Authors have reported and compared the dynamic rheological properties of various starches. Authors have shown the higher value of storage modulus compared to loss modulus in each cases. However, measurement of the damping value, i.e., tan d, can provide important information relating to the viscoelastic behavior and reinforcing effect of fillers.

       7. Please include the conclusion section.

       8. Finally, there are some unwanted grammatical and typos errors                      throughout the entire manuscript.

As a whole, I hope that the authors will take into account all of the aforementioned comments before restructuring their manuscript.    

Reviewer 2 Report

The article reports the physico-chemical properties of the starch of eight common wheat genotypes with different kernel texture.Although the experimental work is very broad and accompanied by a sound statistical analysis,the experimental design is confused and it is not possible to deduce the element of novelty. Furthermore, the application of the study appears to be too restricted to specific local preparations.
In details:
-line 46: Reference n. 3 is not appropriated, it should be removed since it is related only to Indian common wheat cvs
-line 47: Friabilin is not a protein
-line 48: what do you mean by 'major composition'?
-line 50: 'hexaploid'

-line 51: [6] is not appropriate. Substitute it with DOI: 10.1073/pnas.95.11.6262 or other articles by the same author (CF Morris)
-line 69: what do you mean by 'extraordinary soft'? Which are the category of kernel texture by  your method?

-line 91: Jiuqu, in my opinion, is a product of  too marginal and local interest

-line 197: It should be given a correlation between the HI in this work, and the common methods used to determine kernell hardness (i.e. SKCS, PSI, NIR).

-lines 197-198: 'varied in grain hardness ranging' from what? It is not clear

-line 202: 'harder' not 'more grain hardness'

-line 203: 'lower' not 'less'. The sentence is not clear , reformulate it

-lines 218-221: It is a little bit confusing. Why did you isert table 5 here? You should conceive better this statement

-line 228: Which are the waxy varities?

-line 241: please insert a reference here.

-line 378. Insert a dot or a semicolon after 'granule size'

- lines 377-388. The Discussion is really very poor. No relevant conclusions are reported

Round 2

Reviewer 1 Report

The authors have considered most of the issues raised in the earlier review. The current form may be accepted for further processing.

Reviewer 2 Report

I still have some doubts about the importance and setting of the study. The answers given to my comments, in my opinion, did not substantially improve the article